# Genomic Characterization Provides an Insight into the Pathogenicity of the Poplar Canker Bacterium *Lonsdalea populi*

**DOI:** 10.3390/genes12020246

**Published:** 2021-02-09

**Authors:** Xiaomeng Chen, Rui Li, Yonglin Wang, Aining Li

**Affiliations:** Beijing Key Laboratory for Forest Pest Control, College of Forestry, Beijing Forestry University, Beijing 100083, China; cxm498020597@163.com (X.C.); 17801204500@163.com (R.L.); ylwang@bjfu.edu.cn (Y.W.)

**Keywords:** poplar canker, *Lonsdalea populi*, genome, pathogenicity, virulence genes

## Abstract

An emerging poplar canker caused by the gram-negative bacterium, *Lonsdalea populi*, has led to high mortality of hybrid poplars *Populus × euramericana* in China and Europe. The molecular bases of pathogenicity and bark adaptation of *L. populi* have become a focus of recent research. This study revealed the whole genome sequence and identified putative virulence factors of *L. populi*. A high-quality *L. populi* genome sequence was assembled de novo, with a genome size of 3,859,707 bp, containing approximately 3434 genes and 107 RNAs (75 tRNA, 22 rRNA, and 10 ncRNA). The *L. populi* genome contained 380 virulence-associated genes, mainly encoding for adhesion, extracellular enzymes, secretory systems, and two-component transduction systems. The genome had 110 carbohydrate-active enzyme (CAZy)-coding genes and putative secreted proteins. The antibiotic-resistance database annotation listed that *L. populi* was resistant to penicillin, fluoroquinolone, and kasugamycin. Analysis of comparative genomics found that *L. populi* exhibited the highest homology with the *L. britannica* genome and *L. populi* encompassed 1905 specific genes, 1769 dispensable genes, and 1381 conserved genes, suggesting high evolutionary diversity and genomic plasticity. Moreover, the pan genome analysis revealed that the N-5-1 genome is an open genome. These findings provide important resources for understanding the molecular basis of the pathogenicity and biology of *L. populi* and the poplar-bacterium interaction.

## 1. Introduction

Canker diseases of poplar trees are widely distributed in the northern hemisphere and are difficult to manage. Clonal hybrids occupy an important position in poplars, and the improvement of its genetic characteristics makes it have the characteristics of strong resistance and fast growth. *Populus* × *euramericana*, an interspecific hybrid popular of *Populus deltoides* and *Populus nigra*, is one of the most widely grown poplars and is important in landscape greenery and for industrial uses [1,2]. An emerging canker of *P.* × *euramericana* is caused by the gram-negative bacterium, *Lonsdalea populi* (formerly *Lonsdalea quercina* subsp. *populi*), in China and Europe [3,4]. *L. populi* mainly infects stems or branches of 3-year-old *P. × euramericana* in early summer and expands vertically with exuding frothy fluid. Infected poplars may die, and entire branches can be broken off by wind [5]. *L. populi* has also been reported to naturally infect willow trees (*Salix matsudana*) and causes typical canker symptoms [6]. Because the bacterium extensively colonizes poplar bark, chemical measures are ineffective for curing and controlling *Lonsdalea*-infected cankers [5]. A better understanding of the pathogenicity mechanisms is important for developing more effective canker disease control strategies.

Availability of the *L. populi* genome sequence will be helpful in studying its virulence-related genes. Several genes associated with the type III secretion system and two-component systems were identified and characterized from a *L. populi* draft genome sequence [7]. Previous studies showed that KdpE-KdpD affected pathogenicity and positively regulated 44 downstream genes, and DcuS-DsuR was involved in *L. populi* motility, biofilm formation, and virulence [8,9]. However, because the *L. populi* genome sequence remains incomplete, the complexity of these regulatory networks and the gene repertoires of its pathogenesis remain unclear. Next-generation sequencing can enable rapid and high-quality genome sequencing of *L. populi*.

We used the PacBio RS II sequencing platform to assemble and report the high-quality genome sequence of *L. populi* N-5-1. Functional annotation provided information on its pathogenic characteristics, metabolism, and large mobile genetic elements. Through the comparative genomics analysis between *L. populi* N-5-1 and the other representative strains, we further focused on the evolutionary plasticity of *L. populi* N-5-1. In addition, we predicted the mobile genetic elements and secondary metabolites of *L. populi* N-5-1, *Brenneria nigrifluens. DSM.3017*5, *Erwinia amylovora.* CFBP1430, and *Lonsdalea britannica*.477 to disclose their differential mutations during evolution. In this study, the complete *L. populi* N-5-1 genome provides an insight into the molecular basis of *L. populi* pathogenicity and bark colonization, and the comparative genomic analysis was performed to reveal their relationships.

## 2. Materials and Methods

### 2.1. Bacterial Strain and DNA Extraction

*L. populi* N-5-1 was isolated from *P. × euramericana* in Henan, China, in 2006. N-5-1 was cultured overnight in Luria-Bertani liquid at 30 °C to extract the DNA. The overnight culture was collected and rinsed with 2 mL sterile ddH_2_O. Genomic DNA was extracted using the Bacterial Genomic DNA Extraction Kit (Tiangen Biotech, Beijing, China), according to the manufacturer’s protocol. The DNA quality and concentration were determined using a NanoDrop (Thermo Scientific, Waltham, Massachusetts, USA).

### 2.2. Genome Sequencing, Assembly, and Annotation

The N-5-1 genome was sequenced using a PacBio RS II platform and Illumina HiSeq 4000 platform at the Beijing Genomics Institute (Shenzhen, China). Genome assembly and annotation were performed using the GLIMMER3 with Hidden Markov Models, tRNA scan-SE [10], RNAmmer [11], and Rfam [12] databases. The Kyoto Encyclopedia of Genes and Genomes (KEGG), Clusters of Orthologous Groups (COG), NonRedundant Protein, Gene Ontology (GO), Evolutionary Genealogy of Genes: Non-supervised Orthologous Groups, Virulence Factors of Pathogenic Bacteria (VFDB), Antibiotic-Resistance Genes, and Carbohydrate-Active Enzymes (CAZy) databases were used to determine functional categories. Putative secreted N-5-1 proteins were predicted using software, including SignalPv4.1, TMHMMv2.0, and TargetPv2.0 services.

### 2.3. Phylogenetic Analysis

The phylogenetic tree based on the core-pan gene was constructed using the neighbor-joining algorithm with TreeBEST. The average nucleotide identity (ANI) was calculated using the one-to-one and all-to-all modes in JSpeciesWS [13] for all genome sequences in this study. The classification threshold of the ANI index was 95%.

### 2.4. Comparative Genomics

In order to conduct comparative analysis of the N-5-1 genome with the other three relative bacterial genomes (*Brenneria nigrifluens* DSM.30175, *Erwinia amylovora* CFBP1430, and *Lonsdalea britannica* 477), we downloaded the genome sequences from NCBI and performed pan genomes and core genomes analysis using the default setting under the BPGA software package [14]. The synteny of *L. populi* N-5-1 and other strains was determined using MUMmer and BLAST [15]. N-5-1 pan-genomes were clustered using CD-HIT software for rapid clustering of similar proteins. COG cluster analysis was performed on the core genes to predict their functions. Genomic structural collinearity alignment of N-5-1 with the above-mentioned strains was performed and viewed using Mauve software, which identifies regions of sequence homology and assigns colors to different modules.

### 2.5. Nucleotide Sequence Accession Numbers

The complete genome sequences of the chromosome of *L. populi* strain N-5-were submitted to the NCBI GenBank database under the BioProject PRJNA682499, with the accession number CP065534.

## 3. Results

### 3.1. Genomic Characterization of L. populi N-5-1

*L. populi* N-5-1 was sequenced using a PacBio RS II platform in combination with an Illumina HiSeq 4000 platform. The final assembled *L. populi* N-5-1 genome was 3,859,707 bp with a 56.85% GC content (Figure 1; Table 1). We predicted 3327 coding sequences (CDSs) in the genome with an average length of 918 bp per gene. We identified 107 RNAs, including 75 tRNAs, 22 rRNAs, and 10 ncRNAs (Table 1).

Figure 2 shows the general annotation properties of the N-5-1 genome. Among the total predicted gene models, 2786 genes (78.39%) were annotated in the COG database. Most genes were involved in bacterial transportation and metabolism, i.e., amino acid transport and metabolism (13.07%), translation (8.56%), inorganic ion transport, and metabolism (7.72%) and carbohydrate transport and metabolism (7.65%) (Figure 2A). The GO database categorized 2349 genes (66.09%), and enrichment showed that a higher proportion of the genes were involved in biological processes (Figure 2B). The N-5-1 genome contained 110 genes encoding putative CAZymes (Figure 2C). Virulence-related CAZymes were mainly divided into glycosyltransferases (GTs), glycoside hydrolases, and carbohydrate-binding modules. In the N-5-1 genome, six GTs were associated with virulence: GT1, GT2, GT4, GT19, GT26, and GT30, of which, GT2 belongs to the GT-A category; the others belong to the GT-B category. In total, 27 genes encoded virulence factors, mainly serine proteases and carbohydrate-related enzymes (Appendix A).

Furthermore, 2475 genes were assigned to 207 KEGG functional modules enriched in six groups (Figure 2D): “Cellular Processes” (245), “Environmental” (393), “Genetic” (148), “Human Diseases” (116), “Metabolism” (1807), and “Organismal Systems” (34). The N-5-1 genome contained six carbon fixation pathways: the reductive citric acid cycle (M00173/M00620), reductive acetyl-CoA pathway (M00377), 3-hydroxypropionate bicycle (M00376), hydroxy propionate-hydroxybutyrate cycle (M00375), dicarboxylate-hydroxybutyrate cycle (M00374), and the phosphate acetyltransferase-acetate kinase pathway (M00579). In total, 20 genes were associated with antibiotic resistance (Appendix A), including the efflux pump system (i.e., acrA, acrB, mdtH, mdtG, tolC) and the penicillin-resistant PDP gene family (i.e., pbp1a, pbp1b, pbp2). The genome also contained gene products that encoded mediate bacitracin, glycylcycline, kasugamycin, and fluoroquinolone-related proteins.

Using the antiSMASH online software, we predicted secondary metabolite core gene content in genomes of *L. populi* N-5-1, as well as *B. nigrifluens*, *E. amylovora*, and *L. britannica*. As shown in Figure 3, a non-ribosomal peptide synthetase cluster can be identified in each genome. Strikingly, non-ribosomal peptide synthetase clusters were remarkably expanded in the N-5-1 genome. In addition, the results showed that there were nine gene clusters and some homology with known gene clusters (Appendix A). Among them, five predicted gene clusters have homology with the rhizomide gene cluster up to 100%.

To explore the N-5-1 secretome, putative secreted proteins were predicted with Effective T3, SignalP [16], and ProtComp v9.0 [17]. We identified 340 proteins containing typical signal peptide sequences in the N-terminus. Among them, 26 gene sequences located in secretory proteins were predicted in ProtComp V9.0 (Appendix A
Appendix A). Putative secretory systems of *L. populi* were also identified, including T2SS, T3SS, T4SS, and T6SS (Appendix A).

Using Pfam and the HMM model, we predicted and manually verified 54 genes encoding two-component systems, including 29 histidine kinases (HKs) and 25 response regulators (RRs), which included three hybrid HKs, nine orphan HKs, and eight orphan RRs (Table 2). Furthermore, 380 genes were annotated in the VFDB database, including genes for adhesion and invasion, extracellular exoenzymes, the secretion system, iron acquisition, bacterial toxins, and the two-component signal transduction system (Figure 2E; Appendix A). Genes encoding the flagellar proteins, flagellins (flgE/G/I), chemotaxis (cheD), sugar ABC transporter (sugC), and amino acid adenylation (sypC), were also identified.

### 3.2. Phylogenetic Analysis

To demonstrate the phylogenetics of *L. populi* with 12 related bacteria (Table 3), we generated a phylogenetic tree based on single orthologs, which showed that *L. populi* was closer to *Brenneria* and *Dickeya* than to *Erwinia* (Figure 4A). This was consistent with the results of Li and He [6]. The average nucleotide identity (ANI) values of the 13 genomes were counted using JSpecies software. These values ranged from 72.05–99.98% between the strains (Appendix A). The strains were divided into five clades (Figure 4B). The closest ANI value (99.40%) between *L. populi* and the other reference strains was considerably higher than the threshold value of 95–96%. These results revealed an evolutionary relationship between *L. populi* and *Brenneri*.

Phylogenetic analysis verified that the three genera *Lonsdalea*, *Brenneria*, and *Erwinia* belonged to the same ancestor, but developed their own physiological and biochemical characteristics with the adaptation of evolution. Therefore, we investigated the common pathogen of walnut tree canker, *B. nigrifluens*.DSM.30175; *E. amylovora*.CFBP1430, a bacterial pathogen of the genus *Erwinia* that causes worldwide destruction; *L*. *britannica.*477, which has the same pathogen of *Lonsdalea*, and the strain *L. populi* N-5-1 in this study performed comparative genomic analysis to reveal genomic differences during evolution.

### 3.3. Mobile Gene Elements in the N-5-1 Genome

Pathogenic bacterial genomes contain genomic islands (GEIs), which are clusters of genes for toxin production or other pathogenicity factors [18,19]. We identified 18, 11, and 12 GEIs in the N-5-1 genome with SIGI-HMM, IslandPick, and IslandPath-DIMOB software, respectively, and the GEI lengths ranged from 4144–72,581 bp (Table 4). Further analysis showed that all N-5-1 GEIs were assigned different biological functions than other strains. These pathogenicity islands integrated various transporters and transcriptases. Prophage analysis showed that seven prophages were distributed in the N-5-1 genome (Figure 5A). The complete phage, prophage-6, had a total genome length of 43,758 bp and harbored 57 genes (46 of known function; Figure 5B). The sequences between the functional regions were compact, with an average of 750 bp encoding a gene, suggesting that prophage-6 takes full advantage of the entire genome. Transposable element analysis showed that several insertion sequences were distributed over prophage-6 of the N-5-1 genome, including IS3, IS9, IS11, and IS110.

### 3.4. Comparative Genomics of L. populi and Other Relative Bacteria

To explore pan/core genome of N-5-1 and three related bacterial strains, comparative genomic analysis was performed using CD-HIT software. Based on the results of genome-wide clustering, we calculated the relationships among pan genomes, core genomes, and genome number using BPGA software. Figure 6A shows that the fitting equation between pan genome size (f(x)) and genome number (X) is f(x) = 3483.76 × ^0.49109^, the fitting equation between the number of core genes (f1(x)) and the number of genomes (X) is f1(x) = 4727.06e ^0.34619X^, indicating that the number of N-5-1 pangenome increases with the increase of the number of genomes, but the core genomes dwindle. Therefore, it can be inferred that the pan genome of *L. populi* is still open. In addition, 10,123 genes in the pan-genome were clustered into 9264 gene families (Figure 6B and Figure 7B). Among them, 1381 protein-coding genes (1.47%) were referred to as core genes, 1905 were specific, and 4293 were dispensable (Figure 7A). Furthermore, *L. populi* contained 1409 core genes, 1769 dispensable genes, and 376 specific genes. To gain insight into the functional classification of core and dispensable genes in each genome, we performed COG cluster analysis. There were several differences between core-pan genomes in the numbers of genes belonging to the same COG category. For instance, the core genes were enriched in translation, ribosomal structure, and biogenesis (category J); amino acid transport and metabolism (category E) and coenzyme transport and metabolism (category H) (Figure 7C). Similarly, the dispensable genes were distributed to amino acid transport and metabolism (category E); carbohydrate transport and metabolism (category G) and transcription (category K). In general, the core genes and the dispensable genes were related to transport and metabolism. The signal transduction system of *L. populi* is responsible for sensing environmental stimuli and changing the physiological behavior and/or metabolism of bacteria based on these signals. It is noteworthy that the specific genes encoded mobilome: prophages and transposons functions. Transposon-flanking genes accounted for 43.75% of the genes specific to the N-5-1 genome. Transposons are crucial for genome expansion and play an important role in many aspects of genomic evolution, i.e., new gene generation, regulation of gene expression, phylogeny, and genetic diversity assessment. Therefore, we hypothesize that the specific genes, which were located in *L. populi*, played an important role in the evolution of genomes, changes in gene structure, and the development of new genes. Cluster analysis of the gene families of four strains revealed 3682 gene clusters (Figure 6B), among which *L. populi* has a specific gene cluster. In contrast, *B. nigrifluens*.DSM.30175 and *E. pyrifoliae.*Ep1.96 have several specific gene clusters, 25 and 20, respectively, which may be due to the assembly of chromosomes and plasmids in a single file during genome sequencing and assembly. Figure 6C shows the relationship between the increase in the number of new genes and the number of genomes. The synteny of N-5-1 and its related strains was analyzed using Mauve software [20,21]. Mauve analyses showed that N-5-1 exhibited the highest homology with the *L. britannica* genome (Appendix A), suggesting that large-scale interspecies evolution had occurred.

## 4. Discussion

Poplar tree cankers are one of the most destructive diseases of *P. × euramericana* in China and Europe; however, the underlying mechanisms of the pathogenesis are unclear. The genome sequence can be analyzed via bioinformatics to uncover the molecular mechanisms responsible for the pathogenicity and bark adaptation. Here, we reported the complete genome sequence and functional annotation of *L. populi* N-5-1. In contrast to the previously assembled genome sequenced on only an Illumina HiSeq 2000 platform, the current study showed that 12 more genes were identified belonging to the two-component signal transduction systems here than in the study of Yang et al. [9], and the complete genome sequence and genomic features provide insights into its pathogenic mechanism, biocontrol activities, and evolutionary process [22].

Functional annotation and enrichment analysis showed many overrepresented functions, including amino acid transport and metabolism, translation, ribosomal structure, and biogenesis. Moreover, in the KEGG pathway analysis, 69.63% of the total genes were annotated for biosynthesis, including primary and secondary metabolic pathways and antibiotic-resistance regulatory pathways, such as efflux regulation of streptomycin and vancomycin. To compare these four genomes, we found that N-5-1 had a strong ability to synthesize secondary metabolite. Among them, the four strains shared the non-ribosomal peptide synthetase cluster. In addition, N-5-1 could secrete the aryl polyene cluster and siderophore cluster, and the bioactive substance most likely to be secreted is rhizocticin, bicornutin, and luminmide. These secondary metabolites play an important role in the inhibition of pathogens, bacterial motility, biofilm formation, and metabolism.

Comparative genomic analysis enables comprehensively understanding genomic evolutionary diversity [23,24]. Here, the pan-genome of four stains was an open genome (a > 0), showing a diverse evolutionary process and high genetic plasticity [23,25]. The core genome sets accounted for approximately 39.56% of every genome, which is similar to that of *E. coli* (~40%) [26], higher than that of *Bacillus paralicheniformis* (16.6%), and lower than that of *Riemerella anatipestifer* (60.8%) [27]. The core genes were conserved, and whether the core genes reflect virulence or are involved in pathogenicity is uncertain [28,29]. Core-pan genomic COG cluster analysis confirmed that many core genes and dispensable gens were involved in translation and metabolism (Figure 7C). Transposon coding genes accounted for 43.75% of the genes specific to the N-5-1 genome. Specific gene clusters are essential for bacteria to adapt to natural selection and play important roles in their adapting to specific ecological niches. Transposons are crucial for genome expansion and important for many aspects of genomic evolution [30], such as generating new genes, regulating gene expression and phylogeny, and assessing genetic diversity [31].

GEIs were expanded in the N-5-1 genome. Several potentially virulent GEIs were found in the N-5-1 genome, including pathogenicity islands (PAIs) 6, 7, and 8 and resistance GEIs 1, 2, 21, and 39. GI32 contained a complete metal transport system carrying the iron/manganese transporters, SitC. In addition, several GIs encompassed ABC transporter genes essential for bacterial nutrition, virulence, and transport, including PAI21, GI30, and GI34. Compared with N-5-1, the other three strains had single GEI types, most of which were secretory islands encoding secretory system-related proteins. For example, the GEIs of *B. nigrifluens* mainly encode type II toxin-antitoxin system-related proteins and type III secretion system-related proteins. The *E. amylovora* GEIs encode type III secretion system proteins, and the *L. britannica* GEIs are related to the type VI secretion system.

Virulence factor annotation revealed the specific functional characteristics of the strain [32]. The present work showed that N-5-1 contains several virulence factors. To evade host defense mechanisms and complete rapid invasion, pathogens have several flagellar genes [33]. Here, flagellar signaling pathway annotation results indicated that all genes encoding the flagellum were virulence-associated factors. Complete flagellum-driven motor proteins (MotA/B, FliG/M/N) and chemotactic system proteins (CheR/CheB) enabled rapid flagellar movement [34,35].

Lipooligosaccharide (LOS) is the main component of the outer membrane of gram-negative bacteria, which is composed of lipid A and core oligosaccharides [36]. lgtF and rfaF/D are the main components of the LOS structure. lgtF encodes glucose transferase-related proteins, which participates in LOS synthesis, and a loss of RfaF and RfaD shortens the LOS sugar chain. The number of enzymes also positively affects pathogenesis [37]. Most proteins found in N-5-1 CAZymes database were related to GTs. GT enzymes are considered among the most important enzymes in polysaccharide and oligosaccharide synthesis [38]. As per the CAZy database, GT2 and GT4 make up half of these enzymes and play important roles in the GT family [39]. Deeper analysis of the N-5-1 GT enzymes revealed the presence of GT2 and GT4, which have large catalytic C-terminal domains [40]. Secretion systems allow bacteria to invade and harm host plants, and their absence could reduce bacterial virulence [41]. Gram-negative bacteria often exert their toxicity through the secretory system, which transports virulence factors out of the cell, interacts with the host cell, and infects the host [42]. The virulence-related genes contained different types of secretory systems. Three secretory systems, T3SS, T4SS, and T6SS, were distributed in the N-5-1 genome. The *hrp* gene cluster regulates the invasion of pathogenic bacteria in host plants and the hypersensitive necrosis of non-host plants [43]. We identified 15 *hrp* genes in the N-5-1 strains. HrpU and HrpA encoding proteins are homologous to the type III secretion system export apparatus subunit, SctU, and ATP-dependent helicase, HrpA, respectively, in *L. britannica*. VasK in the N-5-1 genome encodes the VI secretion system protein, a putative membrane protein with TssM of *Brenneria* sp. It contains three lcmF domains thought to increase adhesion to epithelial cells, thus increasing the binding frequency [44]. N-5-1 also contains type VI secretion system proteins, and the genes of T6SS deletion may affect the N-5-1 genome iron uptake system, swimming expression, and its pathogenicity, similar to results obtained for *E. amylovor*a [45]. Iron is essential for bacterial metabolism. Bacteria have a unique iron transport system for obtaining necessary iron from the environment [46]. We analyzed iron transport-related genes and demonstrated that 59 genes were involved in pathogenic mechanisms, suggesting that proteins involved in iron transport systems may either directly recognize heme uptake systems or heme-binding proteins and obtain Fe^3+^ (HemA/B/C) from heme or directly recognize it from an iron transporter (SitA/B) or from the Fe^2+^ transport system (FeoA/B) [47]. After obtaining iron, to complete the transport of iron in the cell, N-5-1 releases a unique TonB-dependent transporter, Acr, which assists in membrane transport of substrates [48,49]. Plant pathogenic toxin is a main pathogenic factor among bacteria [50]. It directly controls the host metabolic pathway or indirectly destroys the host defense system [51]. In our study, the N-5-1 genome contained toxin-related genes involved in synergistic actions.

In summary, this study revealed the sequenced genome of the poplar tree canker bacterium, *L. populi*. Genomic analysis of *L. populi* is important for understanding the pathogenic mechanisms of poplar cankers at the bioinformatics level. Future research via gene mutations and protein-level characterization is needed to elucidate the regulatory mechanisms and interactions of the *L. populi* virulence-associated factors.

## Figures and Tables

**Figure 1 genes-12-00246-f001:**
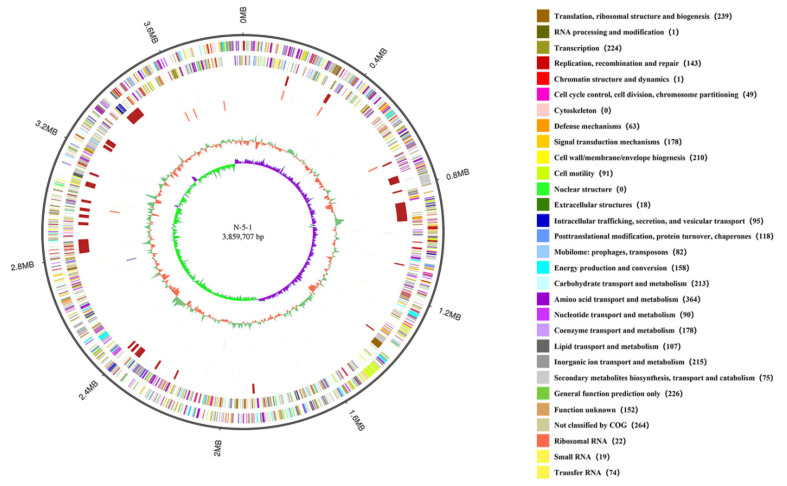
Genome organization of *L. populi* N-5-1. The outermost circle is the coordinate of the genome sequence position. From the outside to the inside, the first circle illustrates the clusters of orthologous groups (COG) annotation gene distribution of the positive strand, colored according to cluster; the second circle depicts the COG annotation of the negative strand; the third circle represents the distribution of the genomic islands by at least one method; the fourth circle denotes the ncRNA distribution of the forwarding strand; the fifth circle represents the ncRNA distribution of the reverse strand; the sixth circle describes the tandem repeat; the seventh circle depicts the GC content, in which the inward red section shows the area within the GC content below the genome-wide average GC content, and the outward green part shows, on the contrary, with the higher the peak of the representation, the greater the difference between the average GC content; the eighth circle denotes the GC skew (G-C/G + C); values >0 in green and values <0 in purple.

**Figure 2 genes-12-00246-f002:**
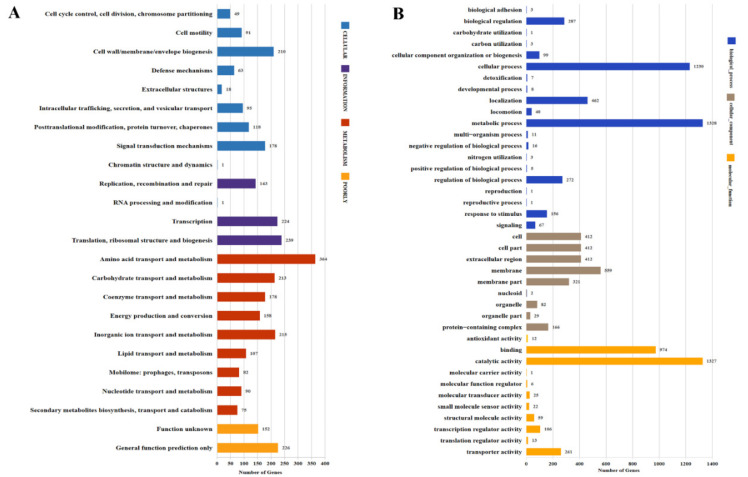
Functional annotation of the *L. populi* N-5-1 genome. Functional classifications were predicted using the COG (**A**), GO (**B**), carbohydrate-active enzyme (CAZy) (**C**), and Kyoto Encyclopedia of Genes and Genomes (KEGG) (**D**) databases. (**E**) According to the Virulence Factors of Pathogenic Bacteria (VFDB) database, the six virulence factor categories were adhesion and invasion, the secretion system, extracellular enzyme, bacterial toxin, the two-component system, and iron acquisition.

**Figure 3 genes-12-00246-f003:**
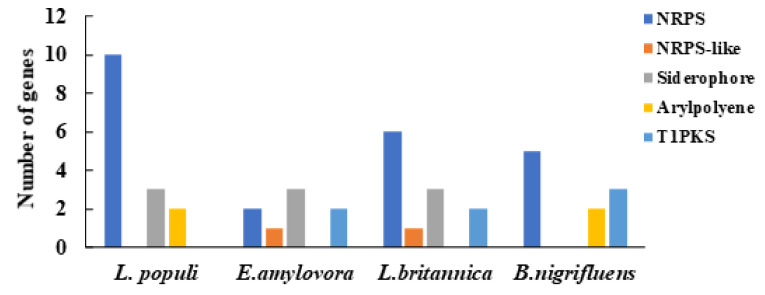
Putative genes encoding secondary metabolites of the four bacterial genomes. Genes involved in the secondary metabolism of the four bacterial genomes (*L. populi* N-5-1, *B. nigrifluens*, *E. amylovora*, and *L. britannica*) were predicted using the antiSMASH software. Non-ribosomal peptide synthetase cluster (NRPS), Siderophore cluster (Siderophore), Arylpolyene cluster (Arylpolyene), and Type I PKS (polyketide synthase) (T1PKS).

**Figure 4 genes-12-00246-f004:**
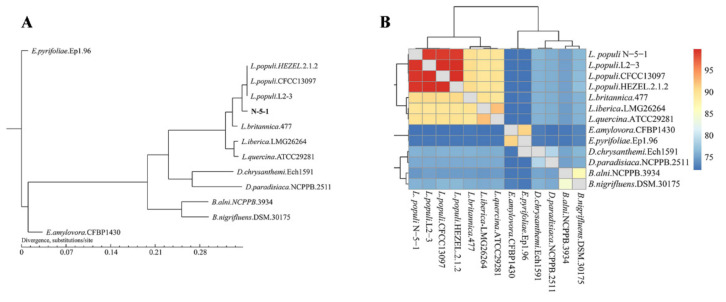
Phylogenetics of *L. populi* N-5-1 and related bacteria. (**A**) Phylogenetic tree constructed using TreeBeST with the neighbor-joining algorithm based on the core-pan gene. Orthologs were predicted via BLASTp-based genome searches. (**B**) Taxonomic thermograms of 13 strains based on the ANI, using online software.

**Figure 5 genes-12-00246-f005:**
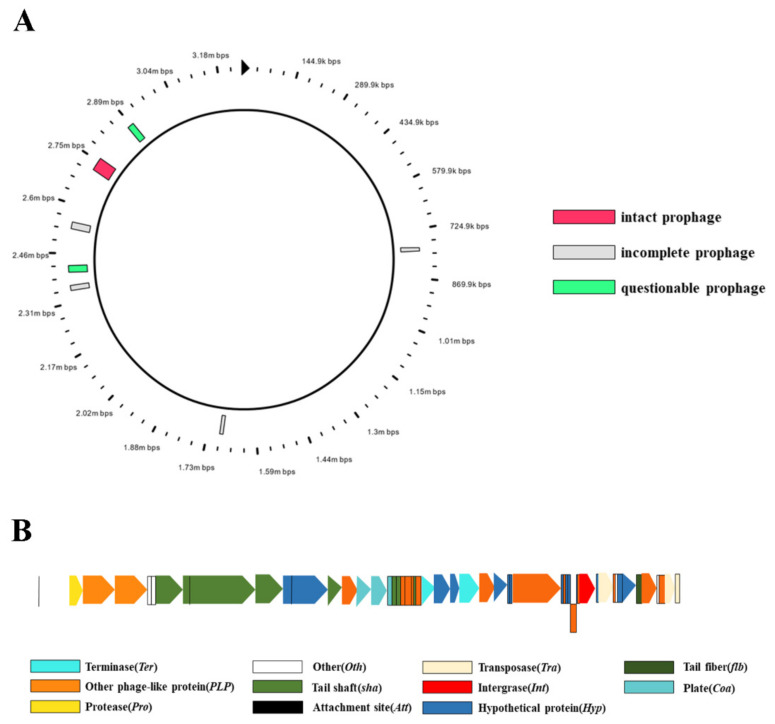
Prophage distribution and structure. (**A**) Prophage distribution in the N-5-1 genome. The red box represents complete prephages; the gray box represents incomplete prephages; the green box represents suspected prephages. (**B**) Prophage-6 structure. Different colored blocks represent genes encoding different functions. Prophage-6 contains 59 coding sequences.

**Figure 6 genes-12-00246-f006:**
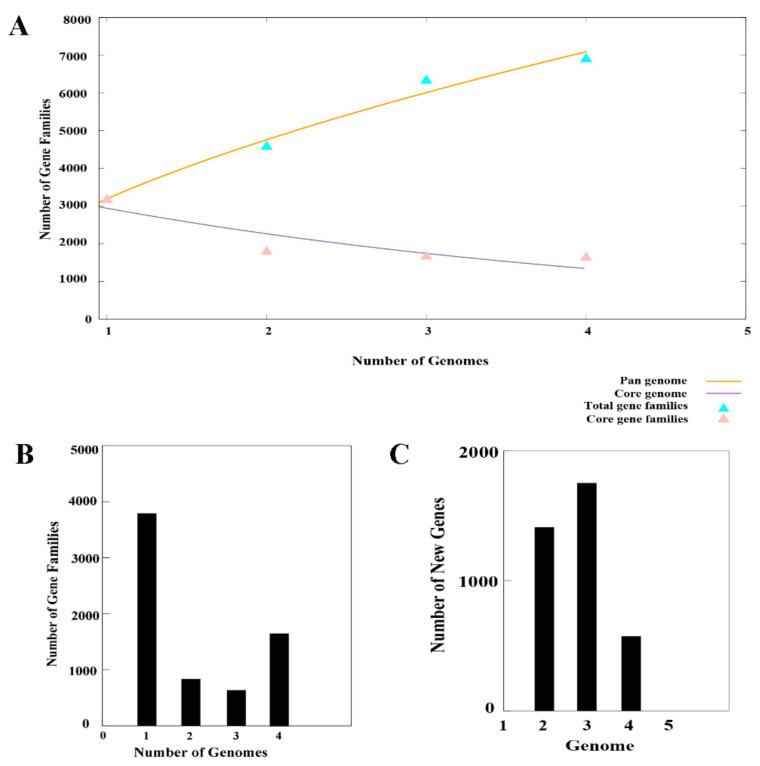
The pan genome of *L. populi* N-5-1 and three closely bacterial genomes. (**A**) Mathematical modeling of pan/core genome of reference strains. The graph shows the size of the pan-genome (orange graphic bars) and size of the core genome (purple bar graphs). (**B**) The bar graph shows genes with varying degrees of conservativeness. (**C**) The number of new genes of four bacterial genomes and the relationship among their genomes.

**Figure 7 genes-12-00246-f007:**
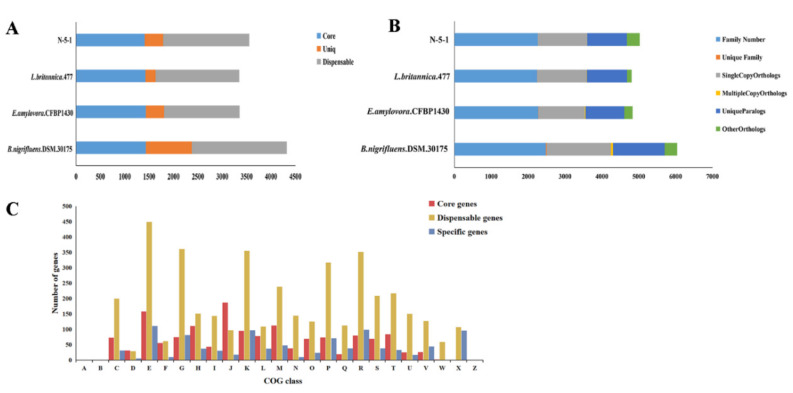
Comparative analysis of *L. populi* N-5-1 with three closely bacterial genomes. (**A**) Chart showing numerous core genes, dispensable genes and specific genes for each of the four stains. These three closely related strains are *B. nigrifluens*, *E. amylovora*, *L. britannica*, and *L. populi* N-5-1, respectively. The different gene types are shown in different colors and sorted according to the assigned number of different genes. (**B**) Numbers of gene families shared by different strains. Different colors represent different homologous genes. Strains are ranked from top to bottom according to the number of different genes. (**C**) Proportion of genes enriched in the clusters of orthologous groups (COG) categories in the core genes, dispensable genes and specific genes according to COG database annotation.

**Table 1 genes-12-00246-t001:** Features of *L. populi* N-5-1 genome.

Features	*L. populi* N-5-1
Genome Size	3859707
GC Content (%)	56.85
CDS ^a^	3554
CDS average Length	917.93
tRNA	74
rRNA (5S rRNA, 16S rRNA, 23S rRNA)	22
Other ncRNA	19

^a^ Coding sequences.

**Table 2 genes-12-00246-t002:** Summary of putative TCS proteins in *L. populi* N-5-1.

Gene Module	HK/RR	Gene Module	HK/RR
**TCS**	GL000251/GL000250	**Hybrid HK**	GL000625
	GL000776/GL000777		GL001047
	GL000952/GL000951		GL003222
	GL001049/GL001048	**Orphan HK**	GL000004
	GL001140/GL001139		GL000382
	GL001547/GL001548		GL001027
	GL001580/GL001581		GL001307
	GL001782/GL001781		GL001885
	GL001852/GL001851		GL001890
	GL001994/GL001993		GL002494
	GL002654/GL002655		GL003203
	GL002081/GL002080	**Orphan RR**	GL000423
	GL003100/GL003101		GL001261
	GL003373/GL003372		GL001275
	GL003423/GL003422		GL001311
	GL003452/GL003453		GL001312
	GL003474/GL003473		GL001708
			GL002169
			GL002492

**Table 3 genes-12-00246-t003:** General genome characters of the 12 associated strains used in this study.

Strains	No. of NCBI Accession	Genome Size (bp)	CDSs	GC%	rRNA	tRNA
*L. populi* N-5-1	NZ_CP065534	3,859,707	3327	56.85	22	75
*L. populi.*HEZEL.2.1.2	NZ_RJUH01000001.1	3,718,244	3100	55.4	10	68
*L. populi.*L2-3	NZ_RJUI01000001.1	3,651,504	3046	55.4	12	68
*L. populi.*CFCC13097	NZ_LUSU01000001.1	3,686,134	3068	55.3	4	61
*L. Britannica.*477	NZ_CP023009.1	4,015,569	3348	55.1	22	83
*L. quercina.*ATCC29281	NZ_FNQS01000023.1	3,779,259	3188	55.6	12	61
*L. iberica.*LMG26264	NZ_LUTP01000001.1	3,781,823	3081	55	5	48
*D. chrysanthemi.*Ech1591	NC_012912.1	4,813,854	4098	54.5	22	74
*B. nigrifluens.*DSM.30175	NZ_CP034036.1	4,891,702	4550	55.9	22	72
*D. paradisiaca.*NCPPB.2511	NZ_CM001857.1	4,631,867	3775	55	22	74
*B. alni.*NCPPB.3934	NZ_MJLZ01000001.1	4,126,956	3768	51	13	65
*E. pyrifoliae.*Ep1.96	NC_012214.1	4,072,846	3502	53.4	22	75
*E. amylovora.*CFBP1430	NC_013961.1	3,833,832	3356	53.6	22	77

**Table 4 genes-12-00246-t004:** Distribution of mobile genetic elements in the N-5-1 genome.

Strain	N.O of GeneIsland	N.O of Prophage
*L. populi* N-5-1	41	7
*B.nigrifluens*.DSM.30175	21	11
*E.amylovora*.CFBP1430	23	4
*L.britannica*.477	8	6

## Data Availability

Data available in publicly accessible repositories. The genome data are e available at http://www.ncbi.nlm.nih.gov/geo/under accession number CP065534.

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
