# Peer review of "Genomic Characterization Provides an Insight into the Pathogenicity of the Poplar Canker Bacterium Lonsdalea populi"

_genes, 2021, doi:10.3390/genes12020246_

Round 1

Reviewer 1 Report

The manuscript Genomic characterization provides an insight into the pathogenicity of the poplar canker bacterium Lonsdalea populi presents very important and new data for Poplar tree cankers. The disease is being one of the most destructive diseases of Poplar species and their hybrids in Europe and China. In this study, the mechanisms of the pathogenesis are underlined.

The genome sequence was analyzed via new methods to understand the molecular mechanisms responsible for the pathogenicity and bark adaptation.

Knowledge of pathogens and bacteria is an interesting scientific subject for forest pathology audience and would be useful for experts working on the same species.

Author Response

   Dear reviewer,

   Thank you for you kind letter of "Genomic characterization provides an insight into the pathogenicity of the poplar canker bacterium Lonsdalea populi "(genes-1096851 ) on 26 Jan 2021. Your postive comments are very helpful for improving our paper.

   Special thanks to you for your good comments.

Reviewer 2 Report

Review for Chen et al. 2021. Genomic characterization Lonsdalea populi.

The main focus of this paper is to describe the genome of L. populi and compare it to several other closely related pathogens, including E. amylovora, B. nigrifluens, and L. Britannica. The authors are able to determine that genome the ration of CAZYmes, tRNAs, and annotate antibiotic-resistance genes within the genome. Overall, the methods and results seem sound, but the writing could be improved throughout the manuscript.

  1. The introduction could be improved to describe objectives that include the genome comparisons conducted.
  2. Also, the description of the core vs. the pan genome could be stronger to highlight methodology and reason for analyzing the pan genome of the one L. populi sequenced.
  3. Figures 1 and 2 text should be larger.
  4. Table 2 could be moved to supplemental.
  5. Table 4 should be rearranged for genetic similarity.
  6. Overall, the content in the discussion is good, but there are numerous editorial mistakes that should be corrected.

Author Response

      Dear reviewer,

   Thank you for you letter and for the reviewer‘s comments concerning our manuscript entitled "Genomic characterization provides an insight into the pathogenicity of the poplar canker bacterium Lonsdalea populi "(genes-1096851 ). Those comments are all valuable and very helpful for revising and improving our paper.We have studied comments carefully and have made correction which we hope meet with approval. Revised portion are marked in red in the attachment.

     Special thanks to you for your comments.

Reviewer 3 Report

I reviewed the paper untitled "Genomic characterization provides an insight into the pathogenicity of the poplar canker bacterium Lonsdalea populi".

This paper describes the genome content of the poplar bacterial genome of Lonsdalea populi. Despite being important to the researchers interested in this disease, the paper is a general description of the genomic features of a species. I would suggest to publish it as a note instead of a full paper. There is not biological story in this paper. Knowing that other Lonsdaela species infecting different hosts have been sequenced, it would have been very interesting to dig the genomic differences among this species an propose factors  involved in poplar infection.

I attached the PDF with my comments

Author Response

       Dear reviewer,

       I’m very grateful to your comments for the manuscript entitled "Genomic characterization provides an insight into the pathogenicity of the poplar canker bacterium Lonsdalea populi "(genes-1096851 ). According with your advice, we have made extensive modification on the original manuscript. Here, we attached revised portion in the format of attachment, for your approval. Revised portion are marked in red in the attachment.

      Special thanks to you for your comments.
